# Cerebellar and subcortical contributions to working memory manipulation
Joshua B. Tan [1,2], Isabella F. Orlando [1], Christopher Whyte [1,2], Annie G. Bryant [1,2,3], Brandon R. Munn [1,2], Giulia Baracchini [1,2], Maedbh King [4], Claire O'Callaghan [1], Eli J. Müller [1,2] & James M. Shine [1,2] ✉

Working memory is critical for manipulating and temporarily storing information during cognitive tasks such as problem-solving. Most models focus primarily on cortical-cortical interactions, neglecting subcortical and cerebellar contributions. Given the extensive connectivity between the cerebellum, subcortex, and cortex, we hypothesize that they contribute distinct, yet complementary, functions during working memory manipulation. To test this, we used functional Magnetic Resonance Imaging (fMRI) to measure blood oxygen-level dependent (BOLD) activity while participants performed a mental rotation task. Our results revealed a distributed network spanning the cortex, subcortex, and cerebellum that differentiates rotated from non-rotated stimuli and correct from incorrect responses. Notably, delayed responses in premotor, subcortical, and cerebellar regions during incorrect trials, suggest that their precise recruitment is crucial for successful working memory manipulation. These findings expand current models of working memory manipulation, revealing the collaborative role of subcortical and cerebellar regions in coordinating higher cognitive functions.

Working memory refers to the ability to store information over a short period of time, such as remembering song lyrics or items from a grocery list[1],[2]. This capacity is thought to involve a distributed network of frontoparietal brain regions that store the information in either local resonant circuits or in short-term plastic inter-synaptic changes[3],[4]. Despite its ubiquity, our understanding of the neural processes governing our capacity to manipulate the information stored in working memory over time remains incomplete[5]. For one, working memory manipulation is a dynamic and highly variable process that involves mentally manipulating and transforming information, clearly distinct from merely storing information[5–9]. It is also less clear how a network of neurons typically associated with unsupervised learning—such as the cerebral cortex—can selectively instantiate the context-dependent, multi-system interactive processes required under the many varied ways in which we manipulate the contents of working memory in our day-to-day lives[10]. Furthermore, it has previously been shown that subcortical structures can provide complementary functions for cognitive processing[10–12]. Here, we advocate for new approaches and perspectives that consider the functional architecture of the whole brain in the mediation of working memory manipulation.

Current popular frameworks characterize working memory as arising from a central system—consisting of brain regions in the prefrontal cortex, parietal lobe and anterior cingulate—that stores and controls information (working memory storage) with accompanying arms that can process and manipulate the information (i.e., a phonological loop in Broca's and Wernicke's area and a visuo-spatial sketch-pad in the occipital lobe)[1,2,5]. While these frameworks provide a foundation for understanding working memory, there is a lack of consensus regarding how information is stored and manipulated[5,13,14]. This is due in part to the complexity of working memory manipulation. Neural signatures underlying working memory manipulation can vary depending on the modality of the information (i.e., visual, verbal, spatial) and the specific manipulation being applied (i.e., continuously updating information as is done when creating a list, or discrete manipulation by mentally transforming information as is done during mental rotation)[5,13]. Furthermore, many of these frameworks are corticocentric, with only a few frameworks including subcortical brain regions, such as the substantia nigra and thalamus[5,14–16]. This has resulted in a lack of discussion and investigation into other regions of the brain, such as the cerebellum, which are consistently recruited in working memory studies but have yet to find a place within these frameworks[17–27]. By continuing to advance analyses and frameworks to be inclusive of regions from the whole brain, a more comprehensive understanding of how the brain processes information can be attained.

There is now strong cross-species evidence that the cerebellum plays a crucial role in working memory[21]. With one of the highest neuronal counts

[1]Brain and Mind Centre, School of Medical Sciences, Faculty of Medicine and Health, University of Sydney, Sydney, Australia. [2]Centre for Complex Systems, The University of Sydney, Sydney, Australia. [3]School of Physics, The University of Sydney, Sydney, Australia. [4]Department of Psychology, University of California, Berkeley, CA, USA. ✉e-mail: mac.shine@sydney.edu.au

in the adult human brain, the cerebellum has a remarkable capacity for information processing[28]. Its modular circuitry[29–32] is linked to distinct cognitive functions, including pattern separation and anticipation[33–37], as well as skill execution and adaptation[33,38–40]. While a common misconception is that the cerebellum's main functions are limited to motor coordination and prediction, evidence indicates that the cerebellum both receives input and sends output back to association areas of the cerebral cortex, such as the prefrontal cortex and parietal lobe[41,42]. Through these connections, the role of the cerebellum extends beyond motor actions to cognitive and multi-domain functions like parallel processing[43].

Building on this broader role, the cerebellum's computational properties make it well-suited for working memory manipulation. Specifically, the cerebellum has been implicated in the formation of internal models that predict how objects will behave under different conditions[33,34,44]. This capacity can then be extended to purely mental scenarios—e.g., in a similar way that the cerebellum helps predict the many possible ways to reach for a cup, it can also predict the possible reconfigurations of mentally rotating an object[44,45]. In this way, the cerebellum may contribute to more abstract concepts, such as action planning and the prediction of outcomes[22,30,46,47], which holds consistently across species[4,46,48]. Furthermore, in studies with impaired cerebellar function (i.e., lesioning, degeneration, optogenetic perturbation), working memory performance is degraded[21,23,24]. In a case where impairments were alleviated through pharmacotherapy, there were changes in the cerebellum that were significantly related to improved performance[49]. Hence, we predict that through the cortico-cerebellar circuitry, the cerebellum contributes significantly to a model of working memory manipulation.

Along with the cerebellum, other subcortical structures—notably, the basal ganglia—serve important roles in working memory manipulation[15]. Currently, the focus on subcortical contributions to working memory is limited to the cortico-basal ganglia circuitry, particularly on the role of dopaminergic signaling in learning[5,14,50]. The basal ganglia play an important role in habituation and goal-directed learning, both important facets for learned skills[51]. Furthermore, the basal ganglia are disynaptically connected with the cerebellum[52,53], suggesting a degree of functional integration that could underpin emergent higher-order functions. Through these connections, the basal ganglia can guide the cerebellum on what to learn[52,53] and thus update the brain's internal model in a way that has the capacity to augment adaptive behavior[11,54]. Therefore, a systems-level perspective that incorporates interactions between the cerebellum, basal ganglia, and cerebral cortical regions can provide a more comprehensive understanding of how the brain manipulates working memory.

In the current study, we analyzed BOLD fMRI patterns within the cerebral cortex, cerebellum, basal ganglia, and thalamus during a mental rotation task, an example of how the brain manipulates working memory. The purpose of this study was to extend previous investigations of working memory manipulation to incorporate spatial–temporal signatures from the whole brain. We hypothesized that successful mental rotation recruits a distributed network of regions that would involve regions from the whole brain. Furthermore, we predicted that the cerebellum and other subcortical regions provide unique contributions to performing the mental rotation task that are not found in the cerebral cortex.

## Results
### Behavioral performance
Twenty-four participants (16 women, 8 men; mean age = 23.8 years, standard deviation (s.d.) = 2.6 years) completed a mental rotation task while undergoing 3 T fMRI scanning (data were analyzed as a sub-sample of a larger battery of tasks)[20]. The stimuli consisted of a pair of three-dimensional objects, with the original object on the left and a target object on the right (Fig. 1a). For each stimulus, there were three possible angles of rotation around the vertical axis that corresponded to differing levels of difficulty: 0° (no rotation, Easy), 50° rotation (Medium), 150° (Hard). Participants were tested across 2 days of scanning, with 8 scans per day (Fig. 1b). Each scan

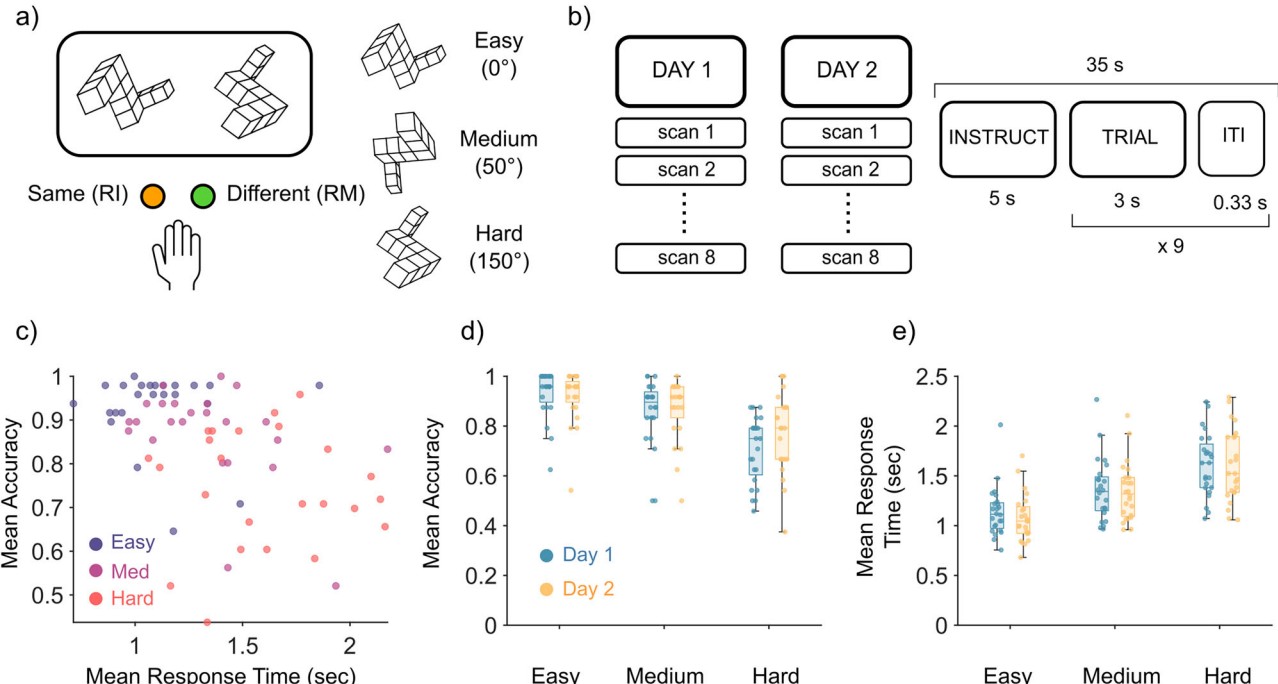

Fig. 1 | Mental rotation task and behavioral results. a Mental rotation stimuli example. Participants respond with two fingers: right middle (RM), right index (RI). Three levels of difficulty based on angles of rotation applied to the stimulus. b Scanning summary with timing of trial. Each participant underwent 2 days of scanning with 8 scans per day. During each scan, the task was performed for 35 s, consisting of 9 trials as laid out. c Scatter plot of mean response time plotted against mean accuracy across all difficulties. Each data point is the mean response time across all trials of that difficulty for a participant. d Mean accuracy across difficulties between each day. Each data point (n = 24) is the mean accuracy of a participant. Center line, median, box limits, upper and lower quartiles. Whiskers, 1.5× interquartile range. e Mean response time across difficulties between days. Each data point (n = 24) is the mean response time of a participant.

consisted of a 35 s mental rotation block including 5 s instruction, 3 s per trial (total 9 trials; 3 per difficulty), and 0.33 s inter-trial interval (ITI). Order of trial difficulty was randomized between scans, with this trial order held consistent across participants. Participants underwent 3 days of training before entering the scanner and were trained on 17 different tasks (refer to King and colleagues[20] for a full list of the tasks). The training program increased in difficulty each day, from focusing on single tasks (day 1) to switching between tasks (day 2), and finally a complete set of the 17 tasks to emulate the protocol used during the scanning sessions. For more details regarding the task paradigm, refer to the "Methods" section. Behavioral performance was measured by accuracy and response time (RT), and only correct trials for these measurements were analyzed. The relationship between accuracy and RT generally followed the speed-accuracy trade-off (Fig. 1c).

We first investigated whether there was an effect of task difficulty on performance. We used generalized linear mixed models to estimate the relationship between task difficulty and testing day with response time and accuracy. When comparing accuracy and task difficulty, we observed a significant effect of task difficulty on accuracy (Fig. 1d). Specifically, for each increase in level of task difficulty, there was a 9% decrease in accuracy ($t_{1149} = -12.89$, $p < 0.001$, CI = -0.12, −0.08). There was no significant relationship between the day of testing and accuracy ($p = 0.55$). When modeling the relationship between response time and task difficulty, a significant relationship was found (Fig. 1e). Specifically, for each increase in level of task difficulty, the response time increased (i.e., responses slowed) by 0.26 s ($t_{2906} = 29.27$, $p < 0.001$, CI = 0.25, 0.28). There was no significant relationship found between the day of testing and response time ($p = 0.12$). From these results, there were differences in behavioral performance across difficulties that were not related to learning across days; hence, the analysis below was collapsed across both days.

## Optimal separation across trial types occurs at distinct timepoints

The fMRI BOLD data were preprocessed and parcellated into 482 regions that covered the cerebral cortex, the cerebellum and a range of subcortical structures[55–58]. We fit a finite impulse response (FIR) model to estimate BOLD activity during a 15 s window per condition (Fig. 2a), allowing us to identify signatures in the BOLD time series related to working memory manipulation. The resulting FIR time series were $z$-scored before undergoing principal component analysis (PCA; Fig. 2b), which provides the principal components (PCs, axes) that capture the most variance of the data, as well as PC loadings, which refer to the original data projected into the PCA space (Fig. 2b). We then trained linear discriminant analysis (LDA) classifiers on the PC loadings to separate out pairs of conditions ($LDA_1$: Easy–Correct vs. Hard–Correct; $LDA_2$: Hard–Correct vs. Hard–Incorrect) at each timepoint. We examined a range of PCs for each timepoint (1–40 PCs), finding that timepoint 6 (with 9 PCs) yielded the best classification for $LDA_1$ and timepoint 15 (with 12 PCs) yielded optimal classification for $LDA_2$. For the final LDA classifiers, we used the first 13 PCs (all PCs > 1% explained variance) which explained a total of ~44% variance of the data (Fig. 2c). By transforming the PC loadings and the PC spatial maps into the LDA space, we can analyze and interpret how the groups were separated, both spatially and temporally. To enforce independence, the LDA axes were normalized and orthogonalized using the Gram-Schmidt procedure. The orthogonalized LDA axes ($LDA_n$) and the original LDA axes ($LDA_1$: Easy–Correct vs. Hard–Correct; $LDA_2$: Hard–Correct vs. Hard–Incorrect) showed a strong linear correlation ($LDA_1$ $|r| = 0.71$, $p = 0.0063$; $LDA_2$ $|r| = 0.70$, $p = 0.0072$). For details on the performance and evaluation of these models, refer to Supplementary Fig. S1. Similar results were replicated using a finer parcellation including 1000 Schaefer cortical regions, 28 cerebellar regions, and 54 subcortical regions (Supplementary Fig. S2).

Distinct distributed networks of regions distinguished the Difficulty level in Correct trials ($LDA_1$: Fig. 2d, e) and Performance on Hard trials ($LDA_2$: Fig. 2f, g)—for a detailed list of the specific atlas labels for these regions, refer to Supplementary Data 1 and 2. Importantly, $LDA_1$ and $LDA_2$

cleanly delineated these axes with an orthogonal basis set. Specifically, $LDA_1$ separated Easy–Correct (purple) from Hard–Correct (pink) trials (Fig. 2e; right panel) in a way that showed characteristic changes over time. Specifically, during timepoints 5–10, Hard trials were easily classifiable, whereas Easy trials were uncertain (i.e., around chance performance). Importantly, even though we did not train the classifiers on data from the Medium condition, the Medium condition trials sat between Easy and Hard trials in the LDA space, underscoring the ability of the LDA classifier to distinguish task difficulty levels (Fig. 2e; right panel). In contrast to $LDA_1$, $LDA_2$ clearly separated Hard-Correct (yellow) and Hard-Incorrect (blue) trials (Fig. 2g; left panel). When looking at the classification across timepoints, the confidence for Hard–Correct increases from $t = 10$ onwards and peaks at $t = 14$, whereas confidence in Hard–Incorrect peaks around $t = 7$ before decreasing (Fig. 2g; right panel). These plots demonstrate that the separation between conditions is dynamic and is dependent on specific timepoints.

## Identifying overlapping regions between task difficulty and performance

To further explore the dynamics of Difficulty ($LDA_1$) and Performance ($LDA_2$), we mapped individual regions into LDA space ($x$-axis: $LDA_1$; $y$-axis: $LDA_2$; Fig. 3a). We then separated regions into four quadrants to identify the regions that were selectively associated with Hard trials and Correct Performance, as we reasoned that this reflected the completion of the task when most challenging, while avoiding non-specific aspects of performance difficulty. In total, we observed a collection of 108 overlapping regions that best differentiated Hard trials that were also performed Correctly (Fig. 3a, orange). Consistent with our original hypothesis, these regions were distributed across the whole brain, including the premotor cortex (bilateral), superior parietal lobe (bilateral), ventral- and orbitofrontal cortex (bilateral), caudate (bilateral), dorso- and ventro-anterior putamen (bilateral), hippocampus (bilateral), nucleus accumbens, and the vermis and lobule VI (bilateral) of the cerebellum (Fig. 3b). For a detailed list of specific atlas labels refer to Supplementary Data 3. Lateralization in these regions was present, with more regions from the left cerebral cortex compared to the right (Fig. 3b). In contrast, regions more classically associated with cognitive control (such as the canonical frontoparietal 'cognitive control network') were found in the quadrant that differentiated Hard from Easy trials, but only when the trials were performed Incorrectly (Supplementary Fig. S3).

We employed a finite impulse response (FIR) model to obtain a time-resolved estimation of BOLD activity, which in turn allowed us to observe the effect of task difficulty within the overlapping regions across time (Fig. 3c–f). Using this approach, we observed a clear increase in BOLD activity around $t = 6$–$10$ s as task difficulty increased, although we also noticed decreased BOLD activity in some regions preceding this window. To quantify these effects, for each overlapping region (per participant) we calculated the area between the curve and the $y$-axis ($y = 0$), which we refer to as the net BOLD response (Fig. 3f). By fitting generalized linear mixed models comparing each region's net BOLD response to increases in task difficulty, we identified regions that increased or decreased in mean net BOLD response due to increasing task difficulty (Fig. 3g; $p_{FDR} < 0.05$). Regions from the premotor, superior parietal lobe, medial extrastriate cortex, vermis and lobule VI of the cerebellum demonstrated increased net BOLD activity as task difficulty increased, whereas regions from the left temporal lobe, left anterior cingulate, and left inferior parietal lobe decreased in net BOLD activity as task difficulty increased.

## Delayed patterns of activity differentiate between correct and incorrect responses

The LDA approach also allowed us to directly compare BOLD dynamics in different trial types, which in turn facilitated unique insights. For instance, comparing the FIR time series between the Hard–Correct and Hard–Incorrect conditions, the amount of BOLD activity did not clearly differentiate the two conditions (Fig. 4a, b). In fact, the BOLD activity during Hard–Correct appears to vary in a specific time window, whereas BOLD activity during Hard–Incorrect varies throughout the whole 15 s period. To

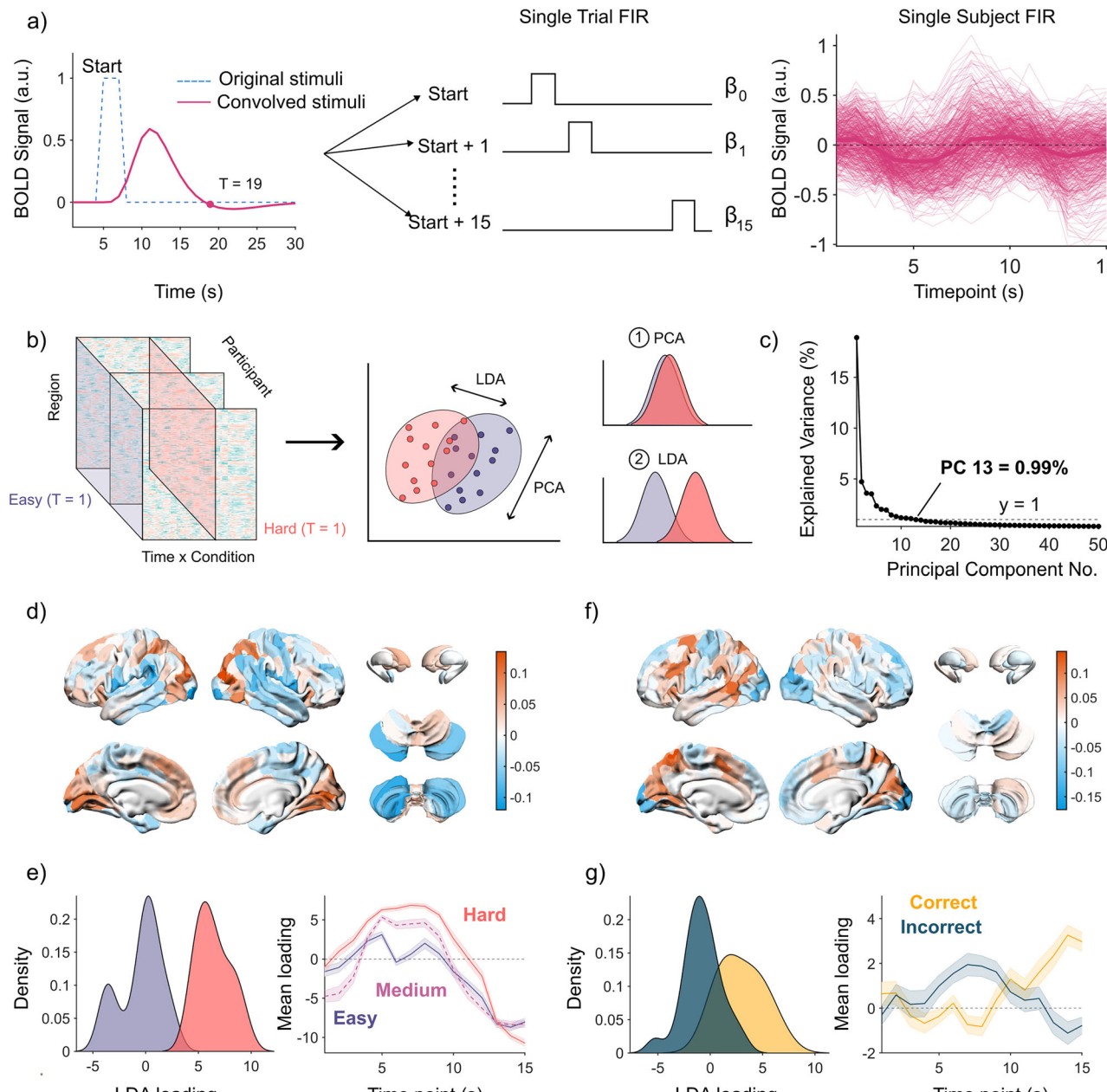

**Fig. 2 | Neuroimaging analysis pipeline. a** Timing of trial (dotted blue line), and estimated timing after convolving with the hemodynamic response function from SPM12 (pink line). Example design matrix, modeling 15 timepoints per condition. Example of the beta coefficients from a single participant using the FIR design matrix, mean BOLD signal colored by a bold line, thinner lines are individual regions ($n = 482$). **b** Overview of dimensionality reduction pipeline. First, the data were z-scored and underwent principal component analysis (PCA). Then the data at each timepoint in PCA space was fed into the LDA classifier with a varying number of principal components. **c** For the final model, we used the first 13 PCs explaining a total of ~44% variance. **d** Regions valued by their ability to separate easy vs. hard trials (orange = useful for Hard, blue = useful for Easy). **e** Left: Distribution of Easy and Hard LDA loadings. Right: Loadings of each difficulty across the duration of a trial. Note: Data were not trained on Medium trials. Mean and standard error are plotted ($n = 24$). **f** Regions valued by their ability to separate Hard–Correct vs. Hard–Incorrect (orange = biased to Hard–Correct, blue = biased to Hard–Incorrect). **g** Left: Distribution of Hard–Correct and Hard–Incorrect LDA loadings. Right: Loadings of each response across the duration of a trial. Mean and standard error are plotted ($n = 24$).

check for changes in variation, we calculated the cross-correlation between corresponding regions of the two conditions. When comparing Hard–Correct and Hard–Incorrect trials for the overlapping regions, we found that the BOLD activity of select regions was delayed during the Hard–Incorrect trials (Fig. 4c; $n = 68$). We next asked whether these differences were unique to performance or also impacted by task difficulty. To this end, we calculated the cross-correlation between Easy and Hard (Fig. 4d) and Medium and Hard (Supplementary Fig. S4) trials. Based on this logic, any increases in the amount of rotation required would result in

temporal variation in BOLD activity across larger windows of time. However, there was no delay effect in this comparison with all regions showing similar timing of BOLD activity (Fig. 4d). The regions that were delayed in Hard–Incorrect trials included the premotor, extrastriate cortex (bilateral), left temporal lobe, left anterior cingulate, left caudate, putamen (bilateral), hippocampus (bilateral), vermis and lobule VI (bilateral) of the cerebellum (Fig. 4e). These results suggest that there were temporal changes driving participants to complete the task successfully, independent of changes due to task difficulty.

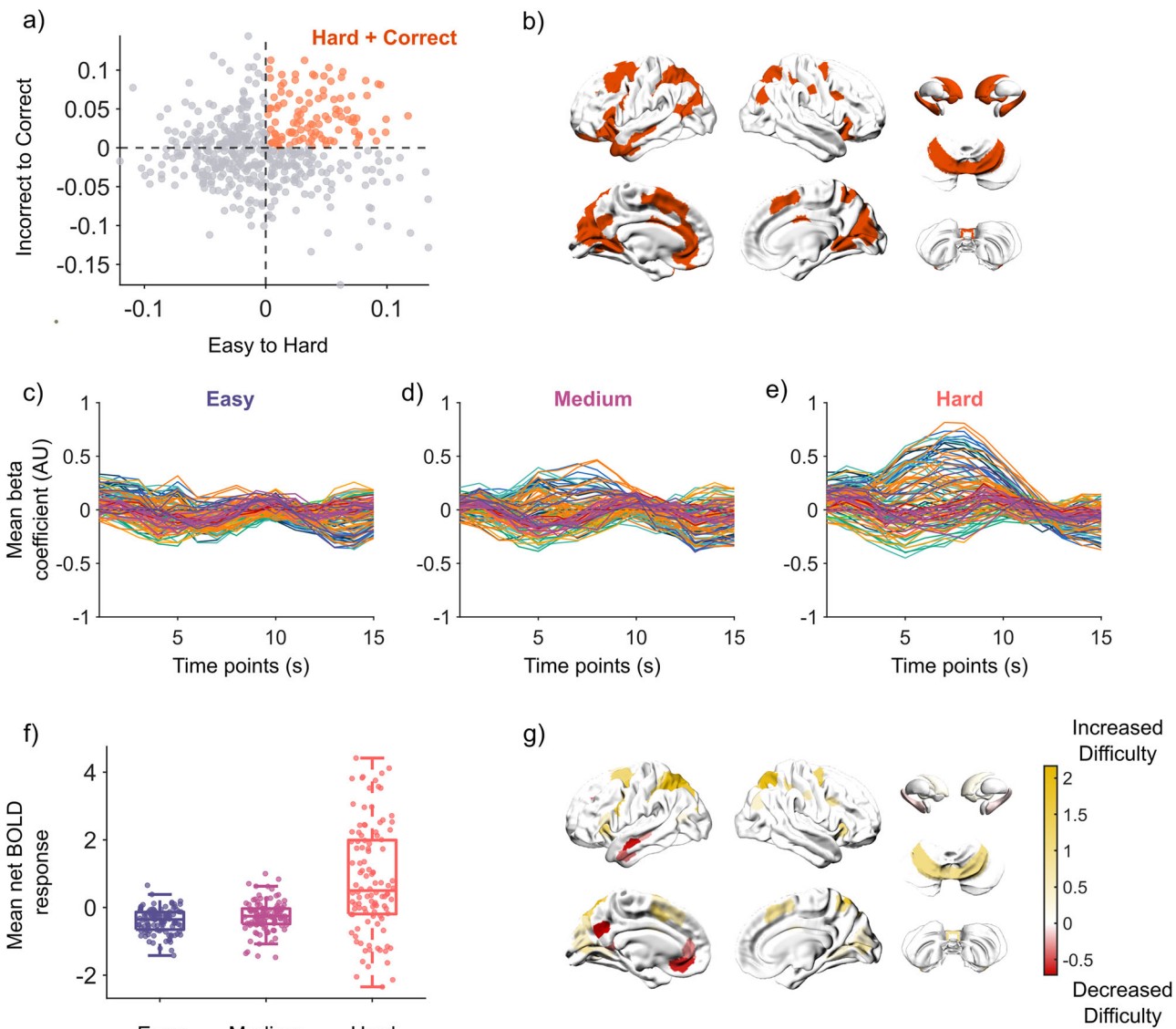

**Fig. 3 | Effect of task difficulty on regional BOLD dynamics. a** Plotting regions by their loadings on both LDA vectors (*n* = 482). **b** Visualizing regions important that separate Hard from Easy, and Hard–Correct from Hard–Incorrect—detailed atlas labels for these regions are in Supplementary data 3 (*n* = 108). For brain visualizations of the other quadrants, refer to Supplementary Fig. S4. **c–e** Group average beta coefficients for **c** Easy, **d** Medium, and **e** Hard trials. Each line is a single region; lines are colored by the 7-Yeo resting-state networks for cerebral regions[58], then the cerebellum, thalamus, hippocampus, basal ganglia, and other subcortical regions as separate groups. **f** Group average net BOLD response per difficulty. Each dot is a region's net BOLD response (*n* = 108). Net BOLD response is calculated by measuring the area between the curve and *y* = 0, where direction does matter (positive and negative values cancel out). Center line, median; box limits, upper and lower quartiles; whiskers, 1.5× interquartile range. **g** Estimated linear relationship between net BOLD and task difficulty as measured from generalized linear mixed models after correcting for multiple comparisons (*n* = 108; false discovery rate *Q* < 0.05).

Another perspective of temporal changes in BOLD activity is attained by observing variance in the BOLD signal, specifically high variance has been linked to cognition, socioemotional measures, and aging[59–61]. One way of measuring variance is through the lens of the energy landscape analysis (refer to the "Methods" section for details on this analysis)[62]. This approach reframes neural time series according to the overall 'state-space' of potential changes that can occur in the data. Through this analysis, we can quantify how much energy (where energy is inferred via the relationship between probability and energy in statistical physics) is associated with a specific brain state (Fig. 4f). By observing the probability of different brain states (high and low BOLD activity), we gain insights into how variable the BOLD signal was across time. To this end, we investigated the probability that a region is recruited or suppressed at each timepoint—measured by the absolute value of BOLD activity. During Hard–Correct trials, between

*t* = 5–10, there is decreased energy requirements (increased probability) for large amounts of BOLD activity to occur (Fig. 4g). Outside of this window, there is a high energy requirement (low probability) of BOLD activity to occur. This contrasts with Hard–Incorrect trials, in which lower energy was required for greater BOLD activity across all timepoints (Fig. 4h). While there was still a slight increase in energy requirements outside of the *t* = 5–10 window, it was lower compared to Hard–Correct trials (Fig. 4h). These results suggest that successful completion of the task is dependent on increasing variance of BOLD activity at specific moments in time.

## Discussion
In the current study, a mental rotation task was used to probe how the brain manipulates working memory. Through a combination of a finite impulse response (FIR) model and linear discriminant analysis (LDA), we identified

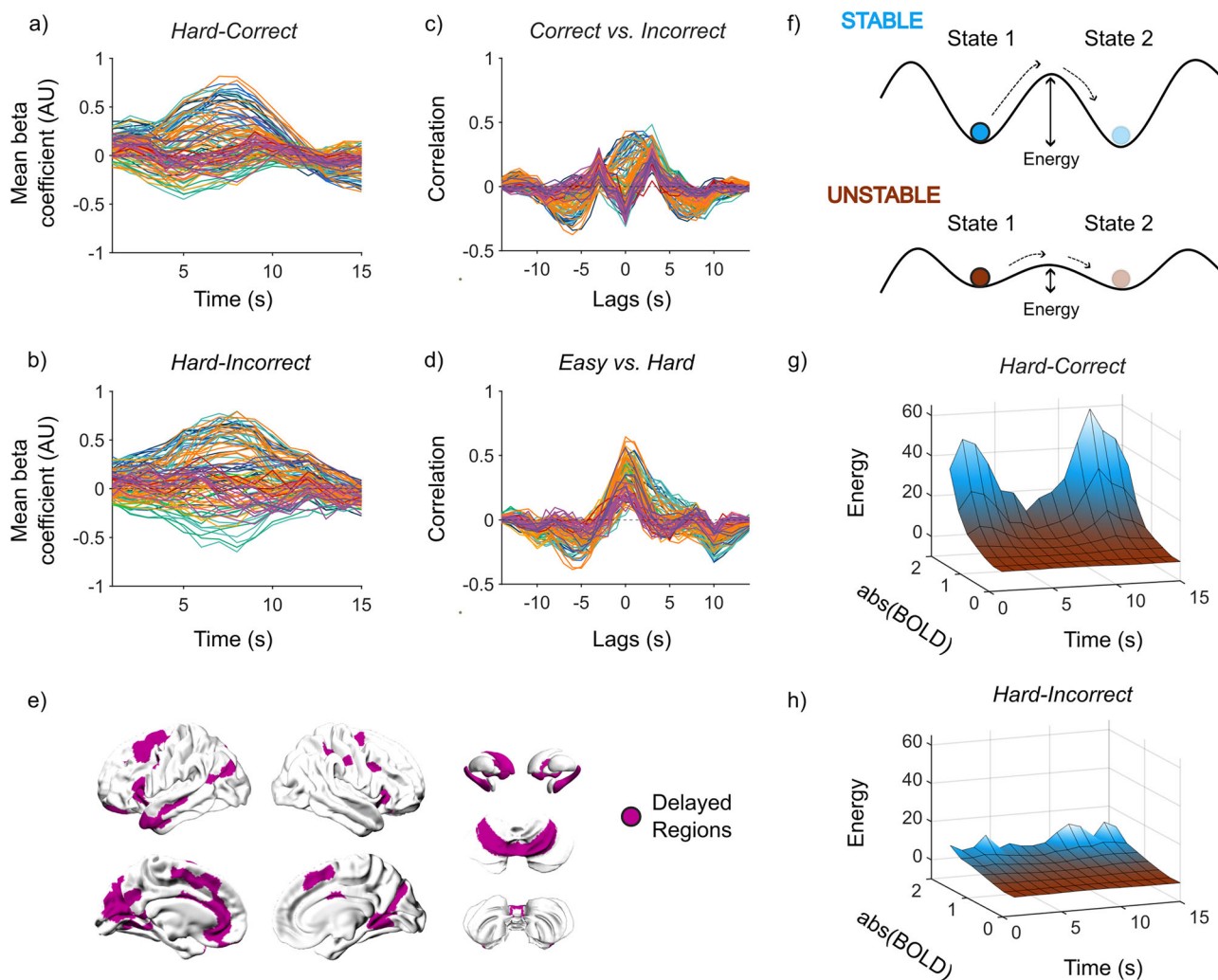

**Fig. 4 | Cross-correlation of corresponding regions between Hard–Correct and Hard–Incorrect. a** Group average beta coefficients for Hard–Correct trials ($n = 108$). **b** Group average Beta coefficients for Hard-Incorrect trials ($n = 108$). **c** Cross-correlation of corresponding regions between Hard–Correct and Hard–Incorrect. **d** Cross-correlation of corresponding regions between Easy and Hard trials. **a–d** Each line represents a single region and is colored by the 7-Yeo resting-state networks for cerebral regions, then cerebellum, thalamus, hippocampus, basal ganglia, and other subcortical regions as separate groups. **e** Regions ($n = 68$) that were delayed by 3 s as seen from (**c**). **f** Schematic describing the intuition behind the attractor landscape analysis. The analysis estimates the probability that an event will occur (i.e. a specific brain state), and this value is then transformed into the energy requirement for the state to occur—greater energy requirements incur a lower probability, while lower energy requirements incur a higher probability. **g** Attractor landscape for the delayed regions in Hard-Correct trials ($n = 68$). **h** Attractor landscape for delayed regions in Hard-Incorrect trials ($n = 68$).

a distributed set of regions spanning the cerebral cortex, cerebellum, and basal ganglia that were important for completing the task correctly during trials of higher difficulty. By probing the temporal dynamics expressed within these regions, we identified unique spatial and temporal characteristics for each condition. Separation between the conditions (Easy vs. Hard, Correct vs. Incorrect) was sensitive to specific timepoints during a trial. Increasing the angle of rotation (increased difficulty) was associated with an overall increase in the BOLD response for regions in the premotor cortex, parietal lobe, and cerebellum, accompanied by decreased BOLD response in the anterior and posterior cingulate cortices and the hippocampus. Hard–Correct and Hard–Incorrect trials showed temporal differences in BOLD response; namely, there was a delayed response in Hard–Incorrect trials compared to Hard–Correct trials in regions from the premotor, medial extrastriate, anterior cingulate, cerebellum, and basal ganglia. Furthermore, the BOLD activity during Hard–Correct trials was more precise—that is, large amplitudes of BOLD activity at specific timepoints, compared to the less constrained BOLD activity during Hard–Incorrect trials. Overall, these results confirm that the manipulation of working memory involves precise recruitment of a distributed network (spanning across the cerebral cortex,

basal ganglia, and cerebellum), and advances our understanding of temporal changes in BOLD recruitment during working memory manipulation.

The notion that manipulation of working memory extends beyond the cerebral cortex has been slowly gaining traction[4,5,15,16,37,46]. However, understanding the specific contributions of each element in such a distributed network is difficult. In this study, by analyzing fMRI BOLD measurements of working memory manipulation, we identified regions beyond the cerebral cortex, such as the vermis, lobule VI of the cerebellum, caudate, putamen, nucleus accumbens, and hippocampus. These regions have been implicated across working memory studies[18,63] but have not been featured in current working memory frameworks[4,5,14]. For example, the vermis and lobule VI of the cerebellum have previously been associated with a range of motor and cognitive functions such as visual working memory, motor planning, and tasks requiring the active maintenance of information[30,37]. These functions are possible due to connections from frontal and posterior parietal areas[41,42], both of which have previously been implicated in non-motor functions such as working memory[64], as well as regions important for task execution, such as the premotor and motor cortices[41]. Furthermore, with the significant role of the cerebellum in learning skills and automaticity[3]

[8,52,54], it is possible that after extensive training in mental rotation (as is the case in the current study), the cerebellum aids in developing a comprehensive model of how to mentally rotate objects[18,37,45]. Hence, proficient working memory manipulation may be achieved through cortico-cerebellar interactions.

Similar to the cerebellum, the basal ganglia are underappreciated in existing working memory manipulation frameworks. The basal ganglia have previously been associated with habituation, reward, and learning processes[51,52], however, much of the discussion around the basal ganglia has focused exclusively on the circuitry involving the thalamus and cerebral cortex[65]. The basal ganglia—specifically, the putamen and caudate (striatum) are also disynaptically connected with the deep cerebellar nuclei[53]. These connections may guide the cerebellum toward specific goals and outcomes[52,53]. Previously, the striatum has been associated with the cortico-cerebellar network during motor planning[38]. The evidence in the present study suggests that not only does cerebellar function generalize for all action planning (both motor and non-motor), but the involvement of the basal ganglia also serves an important role in general action planning. Our results complement and expand on previous findings[17–19,21–27] by linking cerebellar and subcortical activity to behavior, highlighting that recruitment of these regions is important for both task difficulty and correct responses. Therefore, working memory manipulation requires a distributed network that spans beyond the cerebral cortex, involving regions from the cerebellum and basal ganglia.

Expanding current frameworks to include cerebellar and subcortical contributions may have broader implications for treating cognitive function deficits. Mental rotation and working memory manipulation are required for various cognitive functions, which are impaired across diseases and disorders[21,24,66,67]. In some cases, deficits in either the striatum or cerebellum were associated with decreased performance in working memory manipulation[21,24,67]. The current study focused on healthy, young adults. It is likely that there are both similarities and differences in cognitive deficits caused by Parkinson's disease and cerebellar degeneration, and cognitive deficits in a healthy cohort. Therefore, while the results in this study complement working memory literature, future studies should apply similar techniques to investigate neural changes in diseases with working memory deficits.

Execution of working memory manipulation involves both activation and deactivation of specific brain regions. Typically, in task fMRI studies, the focus is on regions that are recruited or increase in BOLD activity during the task condition. However, suppression or inhibition of specific brain regions is also important for task execution[68,69]. In our study, along with the increased recruitment of the regions discussed above, there was a decrease in BOLD activity for regions typically associated with the default mode network and the hippocampus. BOLD activity in the default mode network typically increases during resting-state and has been linked to self-reflection, autobiographical memory, and memory[70,71]. In working memory, the default mode network and hippocampus have both been associated with maintaining information[63],[72]. In contrast to the manipulation of working memory, maintenance is a repetitive process that involves holding information over a period with minimal changes to the information[5,6]. The decreased BOLD activity in the default mode network and hippocampus could be related to the general suppression of these task-negative regions; however, in the context of working memory, these results also point to a difference in neuronal processes between working memory manipulation and working memory maintenance. This would add extra insights into the discussion about maintenance versus manipulation of working memory, which is currently focused on differences in frontal and prefrontal regions[6,7]. Hence, suppression of regions from the default mode network and hippocampus is important for task execution and may distinguish between maintenance and manipulation of working memory processes.

The FIR model teases apart temporal differences in BOLD response, providing a unique perspective on fMRI data. The results from the FIR model and accompanying temporal analyses revealed a temporal precision in regional BOLD time series, which was present only when completing the task correctly. Specifically, there was a distributed network of regions that were delayed in recruitment during hard and incorrect trials. These results demonstrate that the difference between completing the task correctly and incorrectly was driven by timepoint-specific changes, rather than whether a region was recruited or not. In the context of this study, participants were highly trained on the task, such that incorrect responses were more likely to be caused by an unsuccessful completion of the task rather than confusion. This questions the validity of including temporal basis functions in fMRI analyses, and whether recent, more complex models with temporal functions could perhaps tease apart unique temporal changes in the data[73]. Therefore, the use of temporal basis functions in fMRI analysis may provide unique insights into the temporal dynamics of BOLD activity, and future studies could perhaps explore the nuances behind adding different temporal basis functions, finding a balance between minimal constraints and over-fitting parameters to the data.

Proficient working memory manipulation is dependent on the precise recruitment of brain regions. Psychological descriptions of how our brain processes information suggest a continuous relationship between how much an object is rotated and how much rotation our brain processes[74]. For example, for a visual rotation, if the brain is mentally rotating the stimuli (as the task name suggests) to match each other, then the larger the rotation required, the more rotation the brain needs to process. However, from our study, using a basic measure of correlation between Medium and Hard trials, there were no changes in temporal patterns. Furthermore, when comparing Hard-Correct and Hard-Incorrect trials, the recruitment of specific regions was delayed. A possible explanation is that, contrary to intuition, the brain is solving the task by loading a model of mental rotation[33,44,45]. Therefore, under well-practiced conditions, rather than visually rotating the stimuli, participants already know the possible reconfigurations based on previous experience. This coincides with the current results, as the regions presented have previously been associated with action planning and memory[30,37]. Hence, there would be no relationship between angles of rotation and temporal changes, as the approach of this network remains constant regardless of the angle of rotation. Overall, these results expand on previous findings suggesting temporal changes that may influence mental rotation performance.

One key limitation of the current study is the temporal resolution of fMRI. The scanning paradigm used in this study sampled the BOLD signal every second (TR = 1), which may not be sufficient to capture the fast timescale at which mental rotation processes could occur. If mental rotation operates on a sub-second scale, future research employing techniques with higher temporal resolution, such as EEG, could provide valuable insights. The task requires several cognitive steps: stimulus recognition, information processing (mental rotation), and decision-making. Given that participants were well-trained, it is likely that the information processing stage was automated, with response time delays primarily reflecting decision-making[54]. While prior work suggests that increased mental rotation is associated with a positive relationship between response time and task difficulty[13], one limitation of the present work is that we cannot differentiate between the various stages of processing. Future studies could improve this design by incorporating a more nuanced task that isolates these processing stages, allowing for a clearer distinction between automated and controlled cognitive processes while still maintaining some resemblance to the real-world. Other considerations include how the results can generalize across different cohorts (older adults, patient cohorts) and different strategies of working memory manipulation (e.g., reordering or updating information). Working memory deficits in older adults and patient cohorts may be different from deficits observed in a young, healthy cohort like the one in our study. Specifically, it is possible that deficits in older adults and patients may be due to an inability to recruit brain areas that contribute to working memory manipulation[21,23]. Furthermore, neural signatures underlying working memory manipulation can vary depending on the modality (spatial, verbal) and cognitive processes required (manipulating, updating, storage and retrieval)[13]. Overall, questions remain regarding how well the

current results generalize across cohorts and working memory processes, which can be addressed with complementary methods and improvements in task design and data collection.

In summary, our results demonstrate that mental rotation, as an example of working memory manipulation, involves a distributed network consisting of regions beyond the cerebral cortex, such as the cerebellum and subcortex. Current frameworks of working memory manipulation can be biased to the cerebral cortex with minimal discussion about the possible roles of the cerebellum and subcortex. However, these regions are under-appreciated and are associated with working memory manipulation. Using time-resolved model estimations of BOLD activity of regions both from and beyond the cerebral cortex, we found that regions from the cerebellum and subcortex contribute both in terms of the amount of BOLD activity, as well as the temporal patterns during a mental rotation task. These findings suggest an alternative approach to how our brain processes information and underscore the broader role of the cerebellum and subcortex in higher cognitive functions.

## Methods

### Experimental design

The following dataset was originally published by King and colleagues[20] and is openly available for download on OpenNeuro (accession number ds002105). All participants gave informed consent under an experimental protocol approved by the institutional review board at Western University, Canada. All ethical regulations relevant to human research participants were followed. A total of 26 participants participated in this study, with two participants excluded from analyses as they failed to complete all scanning runs. The final sample consisted of 24 healthy individuals (16 women, 8 men; mean age = 23.8 years, s.d. = 2.6) with no self-reported history of neurological or psychiatric illness. All participants self-reported as right-handed (Edinburgh Handedness Inventory > 40).

The original study contained two sets of tasks (sets A and B) of which we only analyzed data from set B. In set B, a total of 17 tasks that covered both motor and cognitive domains were performed during each imaging run, with each task lasting 35 s. Of these 17 tasks, our analyses focused on the mental rotation task which is an example of how the brain manipulates information in working memory[13].

The mental rotation task consisted of stimuli obtained from Ganis and Kievit[74] and were made to follow the traditional Shepard and Metzler three-dimensional stimulus design. The stimuli consisted of a pair of three-dimensional objects: the original object on the left, and a target object on the right. During the task, participants were asked to mentally rotate the target object to determine whether it could be brought into alignment with the original object by only rotating (no flipping/mirroring). There were two types of stimuli: the first being when the two objects could be aligned meaning that they were the "same" object, and the second for when they could not be aligned, in which case they were "different" (Fig. 1a). For each stimulus there were three possible angles of rotation around the vertical axis with differing levels of difficulty: 0° (no rotation, easy), 50° rotation (medium), 150° (hard). These angles of rotation were chosen to reflect most of the behavioral range typically used in mental rotation studies without causing significant self-occlusion at any of the orientations. For more details regarding the presentation and creation of the stimuli, refer to Ganis and Kievit[74].

The mental rotation task was performed across a 35 s block. Within the 35 s, the first 5 s was the instruction period, where the task name ('mental rotation'), the response effector ('use your RIGHT hand') and the button-to-response assignment ('1 = same, 2 = different') were displayed. Following the instruction period, 9 trials of the mental rotation task were presented during the remaining 30 s (duration = 3 s; ITI = 300 ms per trial). The 9 trials included 3 trials per difficulty, with the order of difficulty randomized between scans, but the order was kept consistent across participants. Participants responded using a four-key button box and pressing the buttons with their right index and middle fingers. Feedback was provided in the form of a red X for incorrect or a green tick for correct under the given

stimulus. Cross-session reliability of activation within a task had previously been reported by King and colleagues[20].

### Behavioral training

For tasks in set B, participants completed 3 days of training before the first scanning session. The three training sessions took place over 4–7 days (mean spread of days = 4.4, s.d. = 1.8). During the first day, participants focused on a single task at a time, completing multiple task blocks consecutively. The timing was self-paced, and participants were instructed to read the instructions with the 35 s task block only beginning when they were ready. During this training, online feedback was provided, and an overall accuracy score for the task was provided at the end of the practice.

On the second day of training, switching between tasks was introduced. Participants were given six runs of training, wherein each run consisted of a single task block from 11 different tasks. For the first four runs, the timing was self-paced, like the first day of training. For the final two runs, the instruction period was restricted to 5 s to replicate the timing when in the scanner. On the third day of training, participants practiced 17 tasks in four 10-min runs (35 s per task), emulating the protocol to be used in the scanner sessions.

This training program ensured that participants were familiar with the requirements for each task and had considerable experience in switching between tasks. By introducing training before the scanning, learning during the scanning sessions was minimized. On the third training day, performance was asymptotic, with participants responding correctly to at least 85% of the trials for all the tasks[20].

### Scanning sessions

Participants completed two days of scanning for the tasks in set B. The first day of scanning was conducted within 2–3 days following the final training session (mean = 2.2 days following final training session, s.d. = 1.7 days), and the second day of scanning was completed no more than 7 days after the first day of scanning (mean = 2.7 days following final training session, s.d. = 2.3 days). Each day of scanning consisted of eight imaging runs (10 min per run) with each of the 17 tasks presented once for 35 s per imaging run (Fig. 1b). In total, there were 16 independent measurements of the mental rotation task per participant. The task order was randomized across runs, and to reduce order effects, no two tasks were presented in the same order in two different runs. The order of the runs and the order of the tasks within the run were kept consistent across participants, and novel stimuli were used when possible to reduce the recall of specific stimulus-response associations.

### Imaging acquisition

All fMRI data were acquired on a 3 T Siemens Prisma at the Centre for Functional and Metabolic Mapping at Western University. Whole-brain functional images were acquired using an echo-planar imaging sequence with multi-band acceleration (factor 3, interleaved) and in-plane acceleration (factor 2), developed at the Center for Magnetic Resonance Research at the University of Minnesota. Imaging parameters were as follows: repetition time = 1 s; field-of-view = 20.8 cm; phase encoding direction P to A; 48 slices; 3 mm thickness; in-plane resolution $2.5 \times 2.5 \, \text{mm}^2$. Online physiological recordings of both heart and respiration were acquired during each functional run. For anatomical localization and normalization, a 5 min high-resolution scan of the whole brain was acquired (magnetization-prepared rapid acquisition gradient echo; field-of-view = $15.6 \times 24 \times 24 \, \text{cm}^3$, at $1 \times 1; \times 1 \, \text{mm}^3$ voxel size).

### Pre-processing

Neuroimaging data were organized in BIDS format and pre-processed with fMRIprep version 23.1.4 (https://fmriprep.org/en/stable/#), a standard pipeline that incorporates toolboxes from the gold-standard preprocessing software in the field[75]. fMRIPrep involves the basic pre-processing steps (co-registration, normalization, unwarping, noise component extraction, segmentation, skull-stripping, etc.) and produces a report for quality checking

at each step. See Supplementary Material 1 for a full description of each step. Regression of twelve head motion artifacts, and the average combined white matter, CSF signal was conducted using custom Python scripts, with a high-pass filter set at 0.01. Mean BOLD time series were extracted from 400 Schaefer cortical regions[56], 28 cerebellar regions (SUIT atlas)[55], and 54 sub-cortical regions[57] using custom Python scripts. The following atlases were chosen as robust and popular parcellations that reveal meaningful neuro-biological features[55–57].

## Behavioral analysis

To check whether changes in difficulty affected performance, generalized linear mixed models were constructed using participants' response time (RT) in seconds, and their accuracy (%) across trials within a scan. Only trials that were correctly answered were used for the model of RT. Correct responses were selected as they were likely to be task-related, whereas incorrect responses could be due to several reasons, i.e., making a mistake in the trial, getting distracted, or struggling to handle equipment. Linear mixed models were constructed to find the relationship between RT or accuracy with the level of difficulty (easy, medium, hard) and the specific day on which the scanning took place (Day 1 or 2) while controlling for repeated measurements from the same scan and subject.

## Finite impulse response model (FIR)—the relationship between task difficulty and BOLD activity

Rather than using a single estimate which captures the estimated average amplitude of the response during the task, the finite impulse response (FIR) model allows the estimation of the response at specific time-points of the task. Furthermore, the FIR model is not biased towards a particular shape, such as the hemodynamic response function which is typically used in task-based functional MRI analyses.

A 3 s box-car regressor was convolved with the hemodynamic response function in order to estimate a window length of response related to a single trial (Fig. 2a). From this plot, the convolved curve first dips below zero 14 s after the start of a trial (i.e., 15 frames). Therefore, a window length of 15 s starting at the beginning of the trial and ending 14 s after the start of a trial (15 regressors; 1 per second) was used in the FIR models to estimate the BOLD activity during a trial.

To estimate the relationship between the BOLD response and task difficulty, trials were grouped by difficulty, and the FIR model for each trial was fit to each region's BOLD time series. Only trials with correct responses were included in this model. The final design matrix had 15 regressors (1 per time point) per difficulty ($15 \times 3$), a regressor defining which day the scanning took place, and which scan on the day. Participant-level BOLD activity was estimated by concatenating the BOLD time series for all scans of a participant together and fitting a generalized linear mixed model per participant. In the model, task difficulty ($15 \times 3$ timepoints) and day regressors were modeled as fixed, while the specific scan was controlled for as a random effect. From this model, beta coefficients per timepoint within the 15 s window for each difficulty were obtained. By concatenating trials within each difficulty, temporal blurring between adjacent trials was minimized, and the common effects across similar trials were emphasized.

## Estimating BOLD activity for Hard and Incorrect responses

A similar pipeline was followed to model Incorrect responses during Hard trials. For this comparison, the final design matrix for the FIR model consisted of 15 regressors (15 timepoints for Incorrect responses during Hard trials), a regressor for the day of the scan and a regressor for the specific scan on that day (a total of 32 regressors). Only Hard–Incorrect responses were modeled, as not all participants made errors during Easy and Medium trials.

## Linear discriminant analysis

For each subject, the FIR time series were $z$-scored across regions at each timepoint. The $z$-scored time series underwent principal component analysis (PCA), producing vectors that capture independent axes of variation. The original data were then projected onto these axes of variation and passed through a linear discriminant analysis (LDA) classifier to separate out specific conditions. The initial PCA step minimizes redundant information and enforces independence between variables, improving classification accuracy and robustness[76].

There were two LDAs trained on the data: one separating Easy–Correct from Hard–Correct, and a second separating Hard–Correct from Hard–Incorrect. To have a comprehensive understanding of the classification performance, we trained the LDA across two parameter sweeps: the first across each timepoint ($t = 1$–15), and the second across differing numbers of principal components (max PCs = 40). For each of these classifiers, we evaluated performance using 5-fold nested cross-validation, where folds were constrained to either contain or exclude all measurements of specific participants, as well as balanced accuracy—which is calculated as the average between specificity and sensitivity. To decide which timepoint had the best classification performance, we observed which timepoint had the highest performance (ties included) across varying numbers of PCs. The timepoints that had the best performance for each separation were timepoint 6 (Easy–Correct vs. Hard–Correct) and timepoint 15 (Hard–Correct vs. Hard–Incorrect). The minimum number of PCs to reach the best performance was 9 and 12, respectively. We chose up to the first 13 PCs, accounting for the minimum number of PCs required for both models to reach best performance, as well as following common standard guidelines for using dimensionality reduction methods, in which case we choose up to the first PC which has an added explained variance $< 1\%$[77,78]. The first 13 PCs explained a total of ~44% variance. For a summary of the performance and evaluation of these models, refer to Supplementary Materials 2 and 3.

The resulting two LDA separation axes were trained independently of one another, meaning that these two axes might have shared information. To enforce independence between the two LDA axes, we first normalized each LDA and then orthogonalized the axes following Gram–Schmidt procedures. By definition, as these two axes are now orthogonal, they also describe unique pieces of information from one another. To verify that this transformation has produced axes still related to the original LDA separations, we correlated these new axes ($LDA_n$) to the original LDA axes.

From these two $LDA_n$ separations, we projected the original data onto these axes to see two perspectives: first the contributions of each brain region to the desired separation (spatial), and second how does this separation vary across time and conditions. To check whether these results were consistent across scales, these analyses were also reproduced in a finer parcellation, which consists of 1000 Schaefer cortical regions, 28 cerebellar regions, and 54 subcortical regions.

## Selecting regions important for task performance

From visual observation, the contributions of each brain region to each separation have clear similarities and differences (Fig. 2). Through this process, the two modified maps now define distinct axes for task difficulty and Hard–Correct vs. Hard–Incorrect responses. Each brain region was plotted based on its loadings on the two $LDA_n$ axes (Fig. 3a, b). From this plot, we divided the regions into four quadrants: regions recruited when completing the task correctly and during harder trials (quadrant 1: top right), completing the task correctly and during easier trials (quadrant 2: bottom right), completing the task incorrectly and during easier trials (quadrant 3: bottom left), and completing the task incorrectly during harder trials (quadrant 4: top left) (Fig. 3a). All future analyses only involved regions important for completing the task correctly during harder trials (quadrant 1: top right).

## Estimating the effect of task difficulty on BOLD signal

Using the regions from quadrant 1 (identified previously), we can observe changes in their FIR time series across conditions. To estimate the amplitude of the BOLD signal per region, for each region, we calculated the cumulative area between the time series (curve) and $y = 0$ across all timepoints, where directionality (positive/negative) does matter. In this case, as positive and

negative recruitment will cancel each other out, the total BOLD amplitude of a region from $y = 0$ across the 15 s is the net BOLD amplitude. Generalized linear mixed models were then fit to the data, estimating the effects of task difficulty on the net BOLD amplitude per region, controlling for repeated participants. The resulting estimates were then corrected for multiple comparisons using the false discovery rate (FDR), and significant regions were visualized on the brain.

### Estimating phasic changes in BOLD activity through cross-correlation

When comparing the FIR time series for Hard–Correct and Hard–Incorrect, visually there was no difference between the amplitude of the BOLD signal. To assess whether there were changes in the timing of regional activity, we calculated the cross-correlation of a region's FIR time series between the two conditions. The resulting measurements describe the correlation of a region's activity between both conditions with different delays (lags) in the time series.

### Brain-state displacement and the energy landscape

As previously mentioned, a benefit to using the FIR model design over a traditional box-car is that we can analyze recruitment of a region at specific timepoints of a trial. Building from this idea, we can evaluate the probability that a region is recruited (via increases in BOLD activity) using an approach introduced by Munn and colleagues[62] (code for this analysis is available at https://github.com/ShineLabUSYD/Brainstem_DTI_Attractor_Paper). The energy landscape analysis estimates the probability that a given brain state will take place. This is done by estimating the probability distribution of BOLD activity at each timepoint and inverting this value to estimate the energy required to change from one brain state to another.

In our study, we calculated the energy landscape across the participant-level estimates from the FIR model. The specific conditions we compared were activity during Hard–Correct trials, and Hard-Incorrect trials. The following steps were replicated for these two conditions. For each timepoint, the absolute value of BOLD activity across regions from quadrant 1 was isolated. We then estimated the probability distribution of BOLD activity, $P(\text{BOLD}_t)$, between 0 and 1.8 within the given window using a non-linear Kernel density estimation:

$$P(\text{BOLD}_t) = \frac{1}{0.1n} \sum_{i=1}^{n} K\left(\frac{\text{BOLD}_{t,i}}{0.1}\right)$$

where $K$ is the Gaussian kernel summed over $n$ samples. This probability distribution was then converted to energy, $E$, by taking the negative natural logarithm.

$$E = -ln(P(BOLD_t))$$

From this approach, we can describe the likelihood of time-resolved BOLD activity through the intuition of energy requirements. Highly probable BOLD activity corresponds to a low energy requirement (i.e., small $E$), and an unlikely BOLD activity has higher energy requirements (i.e., large $E$)[62].

### Statistics and reproducibility

Statistics were assessed using a combination of linear mixed models and linear discriminant classifiers. False discovery rate (FDR) correction was applied wherever it was appropriate. Classifiers were trained and assessed using 5-fold nested cross-validation. Behavior and cross-session reliability of activation within a task had previously been reported by King and colleagues[20]. The final sample consisted of 24 healthy individuals (16 women, 8 men; mean age = 23.8 years, s.d. = 2.6). Participants complete two days of scanning. Each day of scanning consisted of eight imaging runs (10 min per run) with each of the 17 tasks presented once for 35 s per imaging run. In total, there were 16 independent measurements of the mental rotation task per participant.

### Reporting summary

Further information on research design is available in the Nature Portfolio Reporting Summary linked to this article.

## Data availability

All imaging and behavioral data are publicly available in an OpenNeuro repository (accession number ds002105).

## Code availability

Analysis of both the behavioral and functional MRI data was conducted in MATLAB v2022b. Code required to reproduce the statistical analyses and figures is publicly available at https://github.com/ShineLabUSYD/WM_Manipulation.

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

## Acknowledgements

The authors acknowledge the University of Sydney HPC service at The University of Sydney for providing HPC resources that have contributed to the processing of the data contained within this data collection. J.B.T. and A.G.B. were funded by the Research Training Program Stipend (SC3227, SC3229). J.M.S. was funded by the Australian Research Council (DP240101295) and the National Health and Medical Research Council (GNT1193857). C.O. was funded by the Australian National Health and Medical Research Council Fellowship (2016866) and the University of Sydney Robinson Fellowship. E.J.M. was funded by the Discovery Early Career Researcher Award (DE250100540). G.B. was funded by the Canadian Institutes of Health Research (MFE193920).

## Author contributions

Conceptualization: J.B.T., J.M.S. Methodology: J.B.T., I.F.O., A.G.B., B.R.M., C.O., E.J.M., J.M.S. Investigation: M.K. Formal analysis: J.B.T. Visualization: J.B.T. Writing—original draft: J.B.T., J.M.S. Writing—review and editing: J.B.T., I.F.O., C.W., A.G.B., B.R.M., G.B., M.K., C.O., E.J.M., J.M.S. All roles and responsibilities of the co-authors were agreed on.

## Competing interests

The authors declare no competing interests.
