## [Transparent Peer Review file · Communications Biology]

Cerebellar and Subcortical Contributions to Working Memory Manipulation

Corresponding Author: Dr James Shine

Version 0:

Reviewer comments:

Reviewer #1

(Remarks to the Author)

In this work, the authors used functional Magnetic Resonance Imaging (fMRI) to investigate changes in brain activity associated with performance of a task involving mental rotation of visually presented geometric objects. The authors used linear discriminant analysis to show that activity on a large network of cortical areas, subcortical structures and cerebellum allowed to differentiate: 1) when the task required mental rotations 2) when a task requiring a large mental rotation was performed correctly.

The work is very interesting and represents an important contribution to the field. The analyses are adequate and support the conclusions made. The work is clearly and thoroughly presented. This said, I think there are a few points which could be improved or made more clear.

1) My main concern with the manuscript is the presentation of the work as an investigation of working-memory. The task used is a mental rotation task, which is not designed to study working-memory. If I understood correctly, both objects are presented simultaneously for 3 seconds, and the task is to judge if the object to the right is or not, the same as the object to the left after rotation of 0, 50 or 150 degrees. If the task was designed with a time interval, where the subject did not see the non-rotated object, then one could distinguish between activity associated to holding object specific information in mind and activity associated with holding and manipulating the information in mind. As the task is designed, I think it is better described as a mental rotation task. As, for example, reading, is usually not described as a task to measure working memory, although it requires working-memory, as many other cognitive tasks.

In the subsection of the methods describing the experimental design it is stated "The mental rotation task was chosen as it has previously been used to test working memory manipulation [Ganis and Kievit]". However, the article cited does not refer to working-memory manipulation, but to mental rotation.

I think the article would benefit by not being framed as an investigation of working-memory, but instead a study of mental-rotation. In my opinion, the relation to working-memory should be discussed in the discussion section, but not be the main frame of the work.

2) If I understood the methods correctly, the way the cross-validation is done, does not take care of the measurements being nested at the subject level. The folds should be done so that all measurements of each subjects are left out. That is the model is trained on some subjects and evaluated on others. Now, the within and between subject level are mixed together, which might result in an over estimation of the performance of the linear discriminant analysis. Alternatively, the whole analysis could be done at the single subject level and then the results could be put together at the subject level in the end. However, this approach might result in other problems, for example having different regions, principal components and time-points allowing classification for different subjects.

3) Figure 2e) shows that Hard trials could be classified on time points 5 to 1, while Easy trials could not. This could be a result of the easy trials requiring very little time to be solved. I think the authors try to address this concern by a cross-correlation analysis. To better address the point, Figure 4 d could show also the cross-correlation between Easy and Hard trials (not only between Medium and Hard).

4) Figure 2g) shows regions with a statistically significant relation between BOLD and task difficulty. The analysis is done on a subset of regions (Figure 2b) which were identified as regions important to separate hard from easy trials (as well as hard-

correct from hard-incorrect trials). I think that the statistical analysis producing the results shown on fig 2g) are not independent on the selection of the regions where the analysis is done. This procedure can result in inflation of the significance of the test statistics. Figure 2 g is still interesting, but the results should not be reported as significant. They could be shown instead as an exploratory analysis instead.

5) The modeling work presented in figures 4 f to h does not add to the manuscript. In the way these results are presented, they do not provide any additional insight to understanding the results already presented. I think they could be removed, or alternatively the benefit of introducing the model should be better justified.

Reviewer #2

(Remarks to the Author)

The manuscript titled "Cerebellar and Subcortical Contributions to Working Memory Manipulation" investigates how subcortical regions, the cerebellum, and cortical areas work together to manipulate working memory. The authors used functional magnetic resonance imaging (fMRI) to analyze BOLD (blood oxygen-level dependent) responses during a mental rotation task. They find that a distributed network involving the cerebellum and subcortical structures is critical for working memory manipulation, challenging traditional models that primarily focus on cortical areas. The study provides new insights into the temporal dynamics and specific brain regions that play a role in the manipulation of working memory.

Overall, the manuscript presents a novel and valuable contribution to the field of cognitive neuroscience. The study's design, particularly the use of fMRI combined with principal component analysis (PCA) and linear discriminant analysis (LDA), is appropriate for addressing the research question. The findings expand our understanding of working memory manipulation beyond the cortex by including subcortical and cerebellar contributions. The paper is well-written and the results are clearly presented. However, there are a few areas where the manuscript could be strengthened, particularly in terms of contextualizing the findings within existing literature and providing more detail on certain aspects of the methodology.

Specific Comments with Recommendations for Addressing Each Comment

1. Introduction Section

Comment 1:

Issue: The introduction outlines the novelty of the study, but it could benefit from a deeper connection to previous fMRI research on the involvement of subcortical and cerebellar regions in working memory. There is a lack of discussion on how this study compares to prior fMRI studies, especially those that also investigate the role of the cerebellum in cognitive functions.

Recommendation: The authors should incorporate more references to fMRI studies that have explored the cerebellum's involvement in working memory, particularly those that focus on the integration of subcortical and cortical regions. They should explicitly state how their findings extend or challenge these existing models.

2. Methods Section

Comment 2:

Issue: The description of the PCA and LDA analysis is somewhat unclear. While the authors provide the technical details, the rationale for choosing specific principal components and the parameters for LDA is not well explained.

Recommendation: The authors should clarify why specific components were retained in the PCA analysis and how the parameters for the LDA model were selected. They should provide more detail on how the number of components was determined, and why these particular components were deemed important for the analysis.

3. Results Section

Comment 3:

Issue: The results regarding the temporal dynamics of BOLD activity are discussed in relation to correct versus incorrect trials. However, the specific role of delayed BOLD responses, particularly during incorrect trials, is not explored in enough detail. The functional significance of these delayed responses remains unclear.

Recommendation: The authors should provide further analysis of the delayed BOLD responses observed during incorrect trials. Specifically, they should explain why certain regions show delayed activation and what this delay means in the context of working memory manipulation and task performance. It would be helpful to link this finding more directly to cognitive processes involved in working memory.

4. Discussion Section

Comment 4:

Issue: While the study offers valuable insights into the role of the cerebellum and subcortical regions in working memory manipulation, the clinical implications of these findings are not sufficiently explored. The potential impact on understanding and treating cognitive disorders related to working memory, such as Parkinson's disease or cerebellar ataxia, is not addressed.

Recommendation: The authors should include a discussion on the clinical implications of their findings. Specifically, they should explore how their results might influence our understanding of cognitive disorders that affect working memory and whether the findings could inform therapeutic strategies for these conditions.

Comment 5:

Issue: The limitations section could be expanded. The authors mention the temporal resolution limits of fMRI but do not discuss other important limitations such as sample size, the diversity of the participants, or the generalizability of the study's findings.

Recommendation: The authors should discuss the limitations of their study in more depth, particularly focusing on the sample size and its potential impact on the generalizability of the results. Additionally, the limitations of using a single task (mental rotation) should be addressed, considering it may not fully capture the complexities of working memory manipulation.

The study is innovative and provides new insights into working memory manipulation, but the authors should address the comments above to enhance the clarity and impact of their findings. The key areas to focus on in revisions include improving the contextualization of the results, clarifying the methodology, discussing limitations more comprehensively, and exploring the clinical implications of the study.

Version 1:

Reviewer comments:

Reviewer #1

(Remarks to the Author)

The authors have satisfactorily addressed all the comments I raised, and I recommend the article for publication.

Reviewer #2

(Remarks to the Author)

I appreciate the authors' thoughtful and comprehensive responses. The manuscript has improved considerably in clarity, methodological transparency, and contextual relevance. Below are my evaluations of the addressed points:

1. The authors have successfully expanded the introduction to include numerous relevant references highlighting prior fMRI research on the cerebellum and subcortical structures in working memory. This strengthens the theoretical framing and clarifies how their findings extend existing models.

2. The methodological explanation regarding PCA and LDA has been improved. The justification for selecting the number of principal components, the use of a parameter sweep, and the cross-validation strategy are now clearly explained and align with best practices for dimensionality reduction and classification.

3. The discussion of delayed BOLD responses in incorrect trials is now more cautious and nuanced. I appreciate the authors' acknowledgment of the temporal limitations of fMRI and their suggestion to use higher-temporal-resolution techniques in future studies. The link to cognitive components such as decision-making and automation is appropriate and well-articulated.

4. The authors have added a balanced discussion on the clinical relevance of their findings. While they appropriately refrain from overgeneralizing to patient populations, the added paragraph connects the current results to cognitive deficits in neurological disorders, which strengthens the translational value of the work.

5. The expanded limitations section now addresses the constraints of the sample (young, healthy adults), task generalizability, and variability in working memory processes across populations and modalities. These additions are welcome and provide a more complete and honest framing of the study's scope.

In summary, the authors have thoroughly and effectively addressed the original concerns. I find the revised manuscript to be significantly improved and support it for publication.

Editor

Please clarify:

- the issue regarding working memory vs. mental rotation

While we agree that the mental rotation task used in this study does not directly modulate the amount of time that the object-to-be-rotated needed to be held over a delay, our use of the term ‘working memory’ in this context referred more directly to the concept of ‘active working memory’ – namely, the work that is performed on information so as to solve a cognitive challenge. We have made this definition more explicit in the manuscript.

- whether you have used a nested-cross validation (if not, please do so)

We have implemented a nested cross-validation, and observed only minor alterations to the results that did not impact our previous conclusions.

- provide more information regarding PCA and LDA analysis.

We have added details to the Methods to help clarify these approaches.

Reviewers' comments:

Reviewer #1 (Remarks to the Author):

In this work, the authors used functional Magnetic Resonance Imaging (fMRI) to investigate changes in brain activity associated with performance of a task involving mental rotation of visually presented geometric objects. The authors used linear discriminant analysis to show that activity on a large network of cortical areas, subcortical structures and cerebellum allowed to differentiate: 1) when the task required mental rotations 2) when a task requiring a large mental rotation was performed correctly.

The work is very interesting and represents an important contribution to the field. The analyses are adequate and support the conclusions made. The work is clearly and thoroughly presented. This said, I think there are a few points which could be improved or made more clear.

Response:

We thank the reviewer for their guidance and agree with the summary of our findings. We have addressed all of the reviewer's concerns below.

1) My main concern with the manuscript is the presentation of the work as an investigation of working-memory. The task used is a mental rotation task, which is not designed to study working-memory. If I understood correctly, both objects are presented simultaneously for 3 seconds, and the task is to judge if the object to the right is or not, the same as the object to the left after rotation of 0, 50 or 150 degrees. If the task was designed with a time interval, where the subject did not see the non-rotated object, then one could distinguish between activity associated to holding object specific information in mind and activity associated with holding and manipulating the information in mind. As the task is designed, I think it is better described as a mental rotation task. As, for example, reading, is usually not described as a task to measure working memory, although it requires working-memory, as many other cognitive tasks.

In the subsection of the methods describing the experimental design it is stated “The mental rotation task was chosen as it has previously been used to test working memory manipulation [Ganis and Kievit].”. However, the article cited does not refer to working-memory manipulation, but to mental rotation.

I think the article would benefit by not being framed as an investigation of working-memory, but instead a study of mental-rotation. In my opinion, the relation to working-memory should be discussed in the discussion section, but not be the main frame of the work.

Response:

The reviewer has pointed out that our task is a mental rotation task and does not measure working memory. First, we agree with the reviewer that the current task does not test working memory by the definition of “holding an item over a delay”. However, we see working memory manipulation as distinct from working memory, where working memory manipulation is a more active process that involves mentally manipulating and transforming maintained information (Wager and Smith, 2003). This distinction has been outlined in numerous previous studies (D’Esposito *et al.*, 1999; Rypma *et al.*, 1999; Basile and Hampton, 2013; Davis *et al.*, 2018). Importantly, within this broader framework, mental rotation is an example of how information can be manipulated (working memory manipulation). For this reason, we have reframed our work around underlying neural signatures of working memory manipulation and not working memory (holding an item over a delay).

In order to improve clarity, we have modified the Introduction to clarify our perspective on working memory manipulation and how mental rotation fits in this framework. To avoid confusion, we have rephrased the Methods section to refer to Mental Rotation as an example of working memory manipulation. We have also added some points in the Discussion, exploring how the current findings generalize to working memory manipulation overall.

Introduction – Pages 3, 5

For one, working memory manipulation is a dynamic and highly variable process *that involves mentally manipulating and transforming information*, clearly distinct from merely storing information (D'Esposito et al., 1999; Nyberg and Eriksson, 2016; Davis et al., 2018; Masse et al., 2019; Chen, Sun and Li, 2023).

While these frameworks provide a foundation for understanding working memory, there is a lack of consensus regarding how information is stored and manipulated (Wager and Smith, 2003; Nyberg and Eriksson, 2016; Nozari and Martin, 2024). *This is due in part to the complexity of working memory manipulation. Neural signatures underlying working memory manipulation can vary depending on the modality of the information (i.e., visual, verbal, spatial) and the specific manipulation being applied (i.e., continuously updating information as is done when creating a list, or discrete manipulation by mentally transforming information as is done during mental rotation)* (Wager and Smith, 2003; Nyberg and Eriksson, 2016).

Page 5

In the current study, we analyzed BOLD fMRI patterns within the cerebral cortex, cerebellum, basal ganglia, and thalamus during a mental rotation task, *an example of how the brain manipulates working memory*. The purpose of this study was to extend previous investigations of working memory manipulation to incorporate spatial-temporal signatures from the whole brain. We hypothesized that successful *mental rotation* recruits a distributed network of regions that would involve regions from the whole brain. Furthermore, we predicted that the cerebellum and other subcortical regions provide unique contributions to performing the mental rotation task that is not found in the cerebral cortex.

Methods – Page 21

Of these 17 tasks, our analyses focused on the mental rotation task, *which is an example of how the brain manipulates information in working memory* (Wager and Smith, 2003).

Discussion – Page 20

Other considerations include how the results can generalize across different cohorts (older adults, patient cohorts) and different strategies of working memory manipulation (e.g. reordering or updating information). Working memory deficits in older adults and patient cohorts may be different to deficits observed in a young, healthy cohort like the one in our study. Specifically, it is possible that deficits in older adults and patients may be due to an inability to recruit brain areas that contribute to working memory manipulation (Ravizza et al., 2006; Devereitt et al., 2019). Furthermore, neural signatures underlying working memory manipulation can vary depending on the modality (spatial, verbal) and cognitive processes required (manipulating, updating, storage and retrieval)(Wager and Smith, 2003). Overall,

questions remain regarding *how well the current results generalize across cohorts and working memory processes* which can be addressed with complementary methods and improvements in task design and data collection.

In summary, our results demonstrate that *mental rotation as an example of working memory manipulation* involves a distributed network consisting of regions beyond the cerebral cortex such as the cerebellum and subcortex. Current frameworks *of working memory manipulation* can be biased to the cerebral cortex with minimal discussion about the possible roles of the cerebellum and subcortex. However, these regions are underappreciated and are associated with working memory *manipulation*.

2) If I understood the methods correctly, the way the cross-validation is done, does not take care of the measurements being nested at the subject level. The folds should be done so that all measurements of each subjects are left out. That is the model is trained on some subjects and evaluated on others. Now, the within and between subject level are mixed together, which might result in an over estimation of the performance of the linear discriminant analysis. Alternatively, the whole analysis could be done at the single subject level and then the results could be put together at the subject level in the end. However, this approach might result in other problems, for example having different regions, principal components and time-points allowing classification for different subjects.

Response:

We agree with the reviewer that the current method does not consider nesting at the subject level and have made changes to the model to account for this. Specifically, we have changed the 5-fold cross-validation to make partitions using subject labels, resulting in folds with all measurements of some subjects being left out. The model was still constructed to discriminate between easy vs hard, and correct vs incorrect.

After rerunning the models across all timepoints with varying number of principal components (PCs), $t = 6$ (with 9 PCs) was best for discriminating between easy and hard, and $t = 15$ (with 12 PCs) was best for discriminating between correct and incorrect which agreed with our original results. There was a minor difference with an extra PC being needed (12 PCs vs 11 PCs originally) for discriminating between correct and incorrect, however we originally used up to the first 13 PCs, therefore this difference has no impact on the results.

We have modified the manuscript and GitHub repository to reflect these changes.

Results – Page 8

We examined a range of PCs for each timepoint (1-45 PCs), finding that timepoint 6 (with 9 PCs) yielded the best classification for LDA₁ and timepoint 15 (with 12 PCs) yielded optimal classification for LDA₂.

Methods – Page 26

For each of these classifiers, we evaluated performance using 5-fold nested cross-validation where folds were constrained to either contain or exclude all measurements of specific participants, as well as balanced accuracy – which is calculated as the average between specificity and sensitivity.

The minimum number of PCs to reach the best performance was 9 and 12 respectively, however we ended up choosing the first 13 PCs.

3) Figure 2e) shows that Hard trials could be classified on time points 5 to 1, while Easy trials could not. This could be a result of the easy trials requiring very little time to be solved. I think the authors try to address this concern by a cross-correlation analysis. To better address the point, Figure 4 d could show also the cross-correlation between Easy and Hard trials (not only between Medium and Hard).

Response:

We agree and have replaced Figure 4d with the cross-correlation between Easy and Hard trials. The cross-correlation between Medium and Hard trials have been moved to the supplementary materials.

The following changes have been made:

Results – Pages 13-14

To this end, we calculated the cross-correlation between Easy and Hard (Figure 4d) and Medium and Hard (Supplementary Materials) trials.

Figure 4 caption. d) Cross-correlation of corresponding regions between *Easy* and *Hard* trials.

Supplementary Materials – Page 34

8. Cross-correlation of Medium vs. Hard trials

Fig. S4

Cross-correlation of Medium vs. Hard trials. Each line is a brain region. Lines are colored by their allocation to the 7-Yeo cortical networks, cerebellum, and subcortex.

4) Figure 2g) shows regions with a statistically significant relation between BOLD and task difficulty. The analysis is done on a subset of regions (Figure 2b) which were identified as regions important to separate hard from easy trials (as well as hard-correct from hard-incorrect trials). I think that the statistical analysis producing the results shown on fig 2g) are not independent on the selection of the regions where the analysis is done. This procedure can result in inflation of the significance of the test statistics. Figure 2 g is still interesting, but the results should not be reported as significant. They could be shown instead as an exploratory analysis instead.

We agree with the reviewer that, through the pipeline we have followed, statistics run on our regions of interest are possibly inflated. As such, we have removed any mentions of significance from the results interpretation. We kept the reference for $p_{FDR} < 0.05$ in order merely to describe how the brain map in Figure 3g has been thresholded.

In the Discussion section (Page 17), we have kept the wording the same as the discussion is focused on how the regions are related to the working memory manipulation and potential insights from considering both increases and decreases in BOLD responses.

Results – Page 12

By fitting generalized linear mixed models comparing each region's net BOLD response to increases in task difficulty, we identified regions that increased or decreased in mean net BOLD response due to increasing task difficulty (Figure 3g; $p_{FDR} < 0.05$). Regions from the premotor, superior parietal lobe, medial extrastriate cortex, vermis and lobule VI of the cerebellum demonstrated increased net BOLD activity as task difficulty increased, whereas regions from the left temporal lobe, left anterior cingulate, and left inferior parietal lobe decreased in net BOLD activity as task difficulty increased.

Discussion – Pages 17-18

Execution of working memory manipulation involves both activation and deactivation of specific brain regions. Typically, in task fMRI studies, the focus is on regions that are recruited or increase in BOLD activity during the task condition. However, suppression or inhibition of specific brain regions is also important for task execution (Constantinidis, Williams and Goldman-Rakic, 2002; Nashef et al., 2022). In our study, along with the increased recruitment of the regions discussed above, there was a decrease in BOLD activity for regions typically associated with the default mode network, and the hippocampus. BOLD activity in the default mode network typically

increases during resting-state and has been linked to self-reflection, autobiographical memory, and memory (Spreng and Grady, 2010; Raichle, 2015). In working memory, the default mode network and hippocampus have both been associated with maintaining information (Čeko et al., 2015; Borders, Ranganath and Yonelinas, 2022). In contrast to the manipulation of working memory, maintenance is a repetitive process that involves holding information over a period with minimal changes to the information (D'Esposito et al., 1999; Nyberg and Eriksson, 2016). The decreased BOLD activity in the default mode network and hippocampus could be related to the general suppression of these task-negative regions; however, in the context of working memory, these results also point to a difference in neuronal processes between working memory manipulation and working memory maintenance. This would add extra insights into the discussion about maintenance versus manipulation of working memory which is currently focused on differences in frontal and prefrontal regions (D'Esposito et al., 1999; Davis et al., 2018). Hence, suppression of regions from the default mode network and hippocampus is important for task execution and may distinguish between maintenance and manipulation of working memory processes.

5) The modeling work presented in figures 4 f to h does not add to the manuscript. In the way these results are presented, they do not provide any additional insight to understanding the results already presented. I think they could be removed, or alternatively the benefit of introducing the model should be better justified.

Response:

The reviewer has mentioned that it is unclear what additional insights are gained by using the energy landscape analysis. The energy landscape analysis is a model of variance and can capture changes in brain states (Munn *et al.*, 2021) which was not explored with the cross-correlation analysis. Variance in fMRI BOLD signal is a useful measure that has previously been linked to cognition, socioemotional measures, and aging (Garrett *et al.*, 2011; Grady and Garrett, 2014; Grady *et al.*, 2023). Linking this all together, we have used the energy landscape analysis to compare variance in the BOLD signal over time during Hard/Correct and Hard/Incorrect trials.

We have made the following changes to justify the use of the model, and the insights gained from using this analysis.

Results – Page 15

Another perspective of temporal changes in BOLD activity is attained by observing variance in the BOLD signal, specifically high variance has been linked to cognition, socioemotional measures, and aging (Garrett et al., 2011; Grady and Garrett, 2014; Grady et al., 2023). One way of measuring variance is through the lens of the energy landscape analysis (refer to Methods for details on this analysis)(Munn et al., 2021).

Through this analysis, we can quantify how much energy (where energy is inferred via the relationship between probability and energy in statistical physics) is associated with a specific brain state (Figure 4f). By observing the probability of different brain states (high and low BOLD activity), we gain insights into how variable the BOLD signal was across time.

These results suggest that successful completion of the task is dependent on increasing variance of BOLD activity at specific moments in time.

Reviewer #2 (Remarks to the Author):

The manuscript titled "Cerebellar and Subcortical Contributions to Working Memory Manipulation" investigates how subcortical regions, the cerebellum, and cortical areas work together to manipulate working memory. The authors used functional magnetic resonance imaging (fMRI) to analyze BOLD (blood oxygen-level dependent) responses during a mental rotation task. They find that a distributed network involving the cerebellum and subcortical structures is critical for working memory manipulation, challenging traditional models that primarily focus on cortical areas. The study provides new insights into the temporal dynamics and specific brain regions that play a role in the manipulation of working memory.

Overall, the manuscript presents a novel and valuable contribution to the field of cognitive neuroscience. The study's design, particularly the use of fMRI combined with principal component analysis (PCA) and linear discriminant analysis (LDA), is appropriate for addressing the research question. The findings expand our understanding of working memory manipulation beyond the cortex by including subcortical and cerebellar contributions. The paper is well-written and the results are clearly presented. However, there are a few areas where the manuscript could be strengthened, particularly in terms of contextualizing the findings within existing literature and providing more detail on certain aspects of the methodology.

Response:

We appreciate the reviewer's comments and are thankful for their detailed feedback. We agree with the reviewer's summary of the key findings and arguments we make in the manuscript. We have addressed all the concerns of the reviewer down below.

Specific Comments with Recommendations for Addressing Each Comment

1. Introduction Section

Comment 1:

Issue: The introduction outlines the novelty of the study, but it could benefit from a deeper connection to previous fMRI research on the involvement of subcortical and cerebellar regions in working memory. There is a lack of discussion on how this study compares to prior fMRI studies, especially those that also investigate the role of the cerebellum in cognitive functions.

Recommendation: The authors should incorporate more references to fMRI studies that have explored the cerebellum's involvement in working memory, particularly those that focus on the integration of subcortical and cortical regions. They should explicitly state how their findings extend or challenge these existing models.

Response:

The reviewer recommended incorporating more references to fMRI studies that link cerebellar activity with working memory, and to explicitly state how our findings relate to previous literature. Regarding including more references, we have added more references to

the appropriate sections in the Introduction. For relating our findings to existing literature, we have added extra points in the Discussion along with the suggestions below, discussing how our findings extend previous existing models.

Introduction – Page 4

This has resulted in a lack of discussion and investigation into other regions of the brain, such as the cerebellum, which are consistently recruited in working memory studies but have yet to find a place within these frameworks (Ravizza et al., 2006; Ben-Yehudah, Guediche and Fiez, 2007; Hayter, Langdon and Ramnani, 2007; Vandervert, Schimpf and Liu, 2007; Hautzel et al., 2009; Tomlinson et al., 2014; Luis et al., 2015; Brissenden and Somers, 2019; Deverett et al., 2019; King et al., 2019; McDougle et al., 2022).

References – Pages 33-34

17. Hautzel, H., Mottaghy, F. M., Specht, K., Müller, H.-W. & Krause, B. J. Evidence of a modality-dependent role of the cerebellum in working memory? An fMRI study comparing verbal and abstract n-back tasks. *NeuroImage* **47**, 2073–2082 (2009).
18. Tomlinson, S. P., Davis, N. J., Morgan, H. M. & Bracewell, R. M. Cerebellar Contributions to Verbal Working Memory. *Cerebellum* **13**, 354–361 (2014).
19. Brissenden, J. A. & Somers, D. C. Cortico-Cerebellar Networks for Visual Attention and Working Memory. *Curr Opin Psychol* **29**, 239–247 (2019).
20. King, M., Hernandez-Castillo, C. R., Poldrack, R. A., Ivry, R. B. & Diedrichsen, J. Functional boundaries in the human cerebellum revealed by a multi-domain task battery. *Nat Neurosci* **22**, 1371–1378 (2019).
21. Deverett, B., Kislin, M., Tank, D. W. & Wang, S. S.-H. Cerebellar disruption impairs working memory during evidence accumulation. *Nat Commun* **10**, 3128 (2019).
22. Vandervert, L. R., Schimpf, P. H. & Liu, H. How Working Memory and the Cerebellum Collaborate to Produce Creativity and Innovation. *Creativity Research Journal* **19**, 1–18 (2007).
23. Ravizza, S. M. et al. Cerebellar damage produces selective deficits in verbal working memory. *Brain* **129**, 306–320 (2006).
24. McDougle, S. D. et al. Continuous manipulation of mental representations is compromised in cerebellar degeneration. *Brain* **145**, 4246–4263 (2022).
25. Hayter, A. L., Langdon, D. W. & Ramnani, N. Cerebellar contributions to working memory. *NeuroImage* **36**, 943–954 (2007).

26. Ben-Yehudah, G., Guediche, S. & Fiez, J. A. Cerebellar contributions to verbal working memory: beyond cognitive theory. *Cerebellum* **6**, 193 (2007).
27. Luis, E. O. et al. Successful Working Memory Processes and Cerebellum in an Elderly Sample: A Neuropsychological and fMRI Study. *PLoS ONE* **10**, e0131536 (2015).

Discussion – Pages 17, 19

Page 17

Our results complement and expand on previous findings (Ravizza et al., 2006; Ben-Yehudah, Guediche and Fiez, 2007; Hayter, Langdon and Ramnani, 2007; Vandervert, Schimpf and Liu, 2007; Hautzel et al., 2009; Tomlinson et al., 2014; Luis et al., 2015; Brissenden and Somers, 2019; Deverett et al., 2019; McDougale et al., 2022) by linking cerebellar and subcortical activity to behaviour, highlighting that recruitment of these regions is important for both task difficulty and correct responses. Therefore, working memory manipulation requires a distributed network that spans beyond the cerebral cortex, involving regions from the cerebellum and basal ganglia.

Page 19

Hence, there would be no relationship between angles of rotation and temporal changes as the approach of this network remains constant regardless of the angle of rotation. Overall, these results expand on previous findings suggesting temporal changes that may influence mental rotation performance.

2. Methods Section

Comment 2:

Issue: The description of the PCA and LDA analysis is somewhat unclear. While the authors provide the technical details, the rationale for choosing specific principal components and the parameters for LDA is not well explained.

Recommendation: The authors should clarify why specific components were retained in the PCA analysis and how the parameters for the LDA model were selected. They should provide more detail on how the number of components was determined, and why these particular components were deemed important for the analysis.

Response:

The rationale of the parameters of the LDA model and why specific components were selected was provided in the Methods section (Page 26). Specifically:

- A parameter sweep was conducted for the LDA model, with 5-fold cross-validation and a combination of model specificity and sensitivity (balanced accuracy) used as measures of model performance
- Parameters were chosen to optimise model performance
- We noted the minimum number of principal components (PCs) to reach maximal performance for both LDA models (9 and 12 respectively) and chose 13 principal components to account for the minimum number of PCs, while following dimensionality reduction guidelines (Nguyen and Holmes, 2019)

We have made the following modifications to the Methods to make these points clear.

Methods – Page 26

For each of these classifiers, we evaluated performance using 5-fold nested cross-validation where folds were constrained to either contain or exclude all measurements of specific participants, as well as balanced accuracy – which is calculated as the average between specificity and sensitivity. To decide which timepoint had the best classification performance, we observed which timepoint had the highest performance (ties included) across varying number of PCs. The timepoints that had the best performance for each separation were timepoint 6 (Easy-Correct vs. Hard-Correct) and timepoint 15 (Hard-Correct vs. Hard-Incorrect). The minimum number of PCs to reach the best performance was 9 and 12 respectively. We chose up to the first 13 PCs, accounting for the minimum number of PCs required for both models to reach best performance as well as following common standard guidelines for using dimensionality reduction methods in which case we choose up to the first PC which has an added explained variance < 1% (Nguyen and Holmes, 2019). The first 13 PCs explained a total of ~44% variance.

3. Results Section

Comment 3:

Issue: The results regarding the temporal dynamics of BOLD activity are discussed in relation to correct versus incorrect trials. However, the specific role of delayed BOLD responses, particularly during incorrect trials, is not explored in enough detail. The functional significance of these delayed responses remains unclear.

Recommendation: The authors should provide further analysis of the delayed BOLD responses observed during incorrect trials. Specifically, they should explain why certain regions show delayed activation and what this delay means in the context of working memory manipulation and task performance. It would be helpful to link this finding more directly to cognitive processes involved in working memory.

Response:

We agree with the reviewer that this result is intriguing and requires further exploration. However, we are cautious of the interpretation in BOLD measurements, due in large part to the inherent limitations in temporal resolution. Specifically, the mismatch in BOLD changes were observed across seconds, whereas the cognitive processes underlying the effective performance of this task occur within a second. Therefore, we raise this limitation in the Discussion section (Page 19) and advocate for further exploration using modalities with higher temporal resolution. In this section we also discuss how the results could relate in the context of working memory manipulation and task performance.

Discussion – Pages 19-20

*One key limitation of the current study is the temporal resolution of fMRI. The scanning paradigm used in this study sampled the BOLD signal every second ($TR = 1$), which may not be sufficient to capture the fast timescale at which mental rotation processes could occur. If mental rotation operates on a sub-second scale, future research employing techniques with higher temporal resolution, such as EEG, could provide valuable insights. The task requires several cognitive steps: stimulus recognition, information processing (mental rotation), and decision-making. Given that participants were well-trained, it is likely that the information processing stage was automated, with response time delays primarily reflecting decision-making (Shine and Shine, 2014). While prior work suggests that increased mental rotation is associated with a positive relationship between response time and task difficulty (Wager and Smith, 2003), one limitation of the present work is that we cannot differentiate between the various stages of processing. Future studies could improve this design by incorporating a more nuanced task that isolates these processing stages, allowing for a clearer distinction between automated and controlled cognitive processes while still maintaining some resemblance to the real-world. **Other considerations include how the results can generalize across different cohorts (older adults, patient cohorts) and different strategies of working memory manipulation (e.g. reordering or updating information). Working memory deficits in older adults and patient cohorts may be different to deficits observed in a young, healthy cohort, like the one used in our study. Specifically, it is possible that deficits in older adults and patients may be due to an inability to recruit brain areas that contribute to working memory manipulation (Ravizza et al., 2006; Devereaux et al., 2019). Furthermore, neural signatures underlying working memory manipulation can vary depending on the modality (spatial, verbal) and cognitive processes required (manipulating, updating, storage and retrieval) (Wager and Smith, 2003). Overall, questions remain regarding how well the current results generalize across cohorts and working memory processes which can be addressed with complementary methods and improvements in task design and data collection.***

4. Discussion Section

Comment 4:

Issue: While the study offers valuable insights into the role of the cerebellum and subcortical regions in working memory manipulation, the clinical implications of these findings are not sufficiently explored. The potential impact on understanding and treating cognitive disorders related to working memory, such as Parkinson's disease or cerebellar ataxia, is not addressed.

Recommendation: The authors should include a discussion on the clinical implications of their findings. Specifically, they should explore how their results might influence our understanding of cognitive disorders that affect working memory and whether the findings could inform therapeutic strategies for these conditions.

Response:

We agree with the reviewer that there are working memory deficits in cognitive disorders and diseases. However, while we can gesture towards how our findings complements the current literature, we are aware that our cohort is different to patient cohorts such as Parkinson's disease and cerebellar ataxia as mentioned by the reviewer. Therefore, making links to therapeutic strategies difficult to infer. We have added a paragraph in the Discussion, mentioning how our findings relate to clinical literature and advocating for future studies to follow a similar pipeline as our study to investigate neural changes in clinical populations.

Discussion – Page 17

Expanding current frameworks to include cerebellar and subcortical contributions may have broader implications for treating cognitive function deficits. Mental rotation and working memory manipulation are required for various cognitive functions which are impaired across diseases and disorders (Park, 1992; Lewis et al., 2003; Deverett et al., 2019; McDougle et al., 2022). In some cases, deficits in either the striatum and cerebellum were associated with decreased performance in working memory manipulation (Lewis et al., 2003; Deverett et al., 2019; McDougle et al., 2022). The current study focused on healthy, young adults. It is likely that there are both similarities and differences in cognitive deficits caused by Parkinson's disease and cerebellar degeneration, and cognitive deficits in a healthy cohort. Therefore, while the results in this study complements working memory literature, future studies should apply similar techniques to investigate neural changes in diseases with working memory deficits.

References – Pages 33, 37

21. Deverett, B., Kislin, M., Tank, D. W. & Wang, S. S.-H. Cerebellar disruption impairs working memory during evidence accumulation. *Nat Commun* **10**, 3128 (2019).
24. McDougle, S. D. et al. Continuous manipulation of mental representations is compromised in cerebellar degeneration. *Brain* **145**, 4246–4263 (2022).

66. Park, S. *Schizophrenics Show Spatial Working Memory Deficits. Arch Gen Psychiatry* **49**, 975 (1992).

67. Lewis, S. J. G. et al. *Using executive heterogeneity to explore the nature of working memory deficits in Parkinson's disease. Neuropsychologia* **41**, 645–654 (2003).

Comment 5:

Issue: The limitations section could be expanded. The authors mention the temporal resolution limits of fMRI but do not discuss other important limitations such as sample size, the diversity of the participants, or the generalizability of the study's findings.

Recommendation: The authors should discuss the limitations of their study in more depth, particularly focusing on the sample size and its potential impact on the generalizability of the results. Additionally, the limitations of using a single task (mental rotation) should be addressed, considering it may not fully capture the complexities of working memory manipulation.

Response:

We have followed the reviewer's advice and expanded the limitations to discuss how the results generalize across populations and tasks. Specifically, how differences in age and healthy vs patient cohorts can change how working memory deficits arise, and that working memory patterns can vary depending on the modality and cognitive process being tested.

Discussion – Page 20

Other considerations include how the results can generalize across different cohorts (older adults, patient cohorts) and different strategies of working memory manipulation (e.g. reordering or updating information). Working memory deficits in older adults and patient cohorts may be different to deficits observed in a young, healthy cohort like the one in our study. Specifically, it is possible that deficits in older adults and patients may be due to an inability to recruit brain areas that contribute to working memory manipulation (Ravizza et al., 2006; Devereaux et al., 2019). Furthermore, neural signatures underlying working memory manipulation can vary depending on the modality (spatial, verbal) and cognitive processes required (manipulating, updating, storage and retrieval) (Wager and Smith, 2003). Overall, questions remain regarding how well the current results generalize across cohorts and working memory processes which can be addressed with complementary methods and improvements in task design and data collection.

The study is innovative and provides new insights into working memory manipulation, but

the authors should address the comments above to enhance the clarity and impact of their findings. The key areas to focus on in revisions include improving the contextualization of the results, clarifying the methodology, discussing limitations more comprehensively, and exploring the clinical implications of the study.

Response:

We thank the reviewer for their detailed feedback and have made appropriate modifications for the sections mentioned.